# Biotin-Decorated PAMAM G4.5 Dendrimer Nanoparticles to Enhance the Delivery, Anti-Proliferative, and Apoptotic Effects of Chemotherapeutic Drug in Cancer Cells

**DOI:** 10.3390/pharmaceutics12050443

**Published:** 2020-05-11

**Authors:** Endiries Yibru Hanurry, Tefera Worku Mekonnen, Abegaz Tizazu Andrgie, Haile Fentahun Darge, Yihenew Simegniew Birhan, Wei-Hsin Hsu, Hsiao-Ying Chou, Chih-Chia Cheng, Juin-Yih Lai, Hsieh-Chih Tsai

**Affiliations:** 1Graduate Institute of Applied Science and Technology, National Taiwan University of Science and Technology, Taipei 106, Taiwan, Republic of China; idrisbrhm7@gmail.com (E.Y.H.); tefe16@gmail.com (T.W.M.); habegaz21@gmail.com (A.T.A.); fentahunhailebdu@gmail.com (H.F.D.); yihenews@gmail.com (Y.S.B.); howlhsu@gmail.com (W.-H.H.); wherelove8@gmail.com (H.-Y.C.); jylai@mail.ntust.edu.tw (J.-Y.L.); 2Advanced Membrane Materials Center, National Taiwan University of Science and Technology, Taipei 106, Taiwan; 3R&D Center for Membrane Technology, Chung Yuan Christian University, Chungli, Taoyuan 320, Taiwan

**Keywords:** biotin, SMVT, gemcitabine, PAMAM dendrimer, anti-proliferation, apoptosis

## Abstract

Biotin receptors are overexpressed by various types of solid cancer cells and play a significant role in tumor metabolism, growth, and metastasis. Thus, targeting the biotin receptors on tumor cells may enhance the efficiency and reduce the side-effects of chemotherapy. The aim of this study was to develop a biotin-coupled poly(amido)amine (PAMAM) (PG4.5) dendrimer nanoparticle to enhance the tumor-specific delivery and intracellular uptake of anticancer drugs via receptor-mediated endocytosis. We modified PG4.5 with diethylenetriamine (DETA) followed by biotin via an amide bond and characterized the resulting PG4.5-DETA-biotin nanoparticles by ^1^H NMR, FTIR, and Raman spectroscopy. Loading and releasing of gemcitabine (GEM) from PG4.5-DETA-biotin were evaluated by UV–Visible spectrophotometry. Cell viability and cellular uptake were examined by MTT assay and flow cytometry to assess the biocompatibility, cellular internalization efficiency and antiproliferative activity of PG4.5-DETA-biotin/GEM. Gemcitabine-loaded PG4.5-DETA-biotin nanoparticles were spherical with a particle size of 81.6 ± 6.08 nm and zeta potential of 0.47 ± 1.25 mV. Maximum drug-loading content and encapsulation efficiency were 10.84 ± 0.16% and 47.01 ± 0.71%, respectively. Nearly 60.54 ± 1.99% and 73.96 ± 1.14% of gemcitabine was released from PG4.5-DETA-biotin/GEM nanoparticles after 48 h at the acidic pH values of 6.5 and 5, respectively. Flow cytometry and fluorescence microscopy of cellular uptake results revealed PG4.5-DETA-biotin/GEM nanoparticles selectively targeted cancer cells in vitro. Cytotoxicity assays demonstrated gemcitabine-loaded PG4.5-DETA-biotin significantly reduced cell viability and induced apoptosis in HeLa cells. Thus, biotin-coupled PG4.5-DETA nanocarrier could provide an effective, targeted drug delivery system and selectively convey gemcitabine into tumor cells.

## 1. Introduction

Cancer is a heterogeneous disease characterized by overexpression of oncogenes that lead to uncontrolled cell division [1]. Various strategies, including immunotherapy and gene therapy, have emerged as novel advanced approaches to treat cancer, conventional chemotherapeutic approaches still represent the cornerstone of cancer treatment [2]. Most chemotherapeutic drugs have common characteristics. However, gemcitabine has unique pharmacological properties and advantage such as its potent cytostatic and cytotoxic activity along with manageable toxicities [3]. Unlike other chemotherapeutic drugs to inhibit polymerase enzymes, gemcitabine can incorporate itself into DNA by replacing cytidine to trigger anti proliferative activity [4]. Furthermore, it is a deoxycytidine analogue of a gold-standard chemotherapeutic drug for unfit and elderly patient of advanced pancreatic, breast, lung, colon, bladder, cervical, and ovarian cancer [5,6]. Gemcitabine diffuses into the cell nucleus where it incorporates or binds to DNA polymerase and blocks the G1/S1 phase of the cell cycle and induces anti-proliferative effects and apoptosis [7,8]. Despite its low molecular weight and lipophobic nature, gemcitabine undergoes extensive first-pass metabolism and is rapidly deaminated in the blood, thus the plasma concentration of the drug drops below effective levels before passing through the whole body [9,10,11].

Therefore, administration of high doses of gemcitabine is required to achieve maximum anti-proliferative activity. However, high doses lead to drug toxicity, drug resistance, and side-effects in normal healthy tissues [12,13,14]. The development of drug nanocarriers to overcome these limitations on the efficacy of gemcitabine has attracted significant attention [15]. Ideal nanocarriers are required to retain and selectively release the drug specifically in tumor cells and need to be compatible with the immune system and exhibit a long lifespan in the circulation [7,16]. A comprehensive range of nanocarriers with different nanostructures have been fabricated to enhance the therapeutic efficiency of gemcitabine, including polymersomes [17], liposomes [18], micelles [14,19], and polymeric nanoparticles, including dendrimers [15,20,21].

Dendrimers are the three-dimensional, hyper-branched macromolecules containing core, void space, and reactive surface regions [22,23]. The poly(amido)amine (PAMAM) dendrimer has been extensively explored as a gene and drug carrier system [24,25]; its advantages of simple fabrication, nanometer size, non-immunogenicity, water solubility, biodegradability, and biocompatibility make the PAMAM dendrimer a suitable synthetic polymer for drug delivery systems [26]. The uniformity and hyper-branched characteristics of PAMAM enable the dendrimer to cross the cell membrane of cancer cells and carry a high cargo content [27]. The low pKa value of the PAMAM dendrimer reduces nonspecific binding of blood proteins and enables sustained release of the drug through the protonation of dendrimer [21]. Several reports have demonstrated the core and void space of PAMAM dendrimer can encapsulate chemotherapeutic drugs and mediate excellent cellular drug delivery [20]. Similarly, the high reactivity of the surface groups enable modification of the PAMAM dendrimer with different active targeting groups. Several active targeting ligands including monoclonal antibodies [16,28], peptides [29], transferrin, folic acid, and other vitamins [27,30,31] have been coupled to dendrimers to boost their cellular uptake and increase their efficiency as drug nanocarriers.

Biotin is essential for normal cell growth and serves as co-enzyme for carboxylase enzymes in the synthesis of fatty acids and branched chain amino acids [32,33]. Solid tumor cells require high levels of biotin due to their high metabolic activity [34]. Biotin receptors are overexpressed on the surface of many tumor cells and thus represent a biomarker for targeting tumor cells [35,36,37,38]. Due to this properties, biotin has emerged as a prominent active targeting moiety for drug nanocarrier systems. Biotin-coupled nanocarriers were reported to demonstrate higher tissue specificity and enhanced cellular uptake via receptor-mediated endocytosis [39,40]. In receptor mediated endocytosis, the transporters present on the surface of cell membrane further boost the uptake and targeting efficiency [41]. Biotin targeted nanocarrier system primarily require sodium dependent multivitamin transporter (SMVT) for endocytosis to encourage selective targeting of nanoparticles [33,42]. SMVT targeted nanocarrier has become a powerful method to deliver hydrophilic drug which cannot easily cross cell membrane [43]. Moreover, the low molecular weight of biotin reduces steric hindrance during reaction and its imidazolidone group also enable the formation of stable nanocarriers via hydrogen bonding [44]. 

In the present study, we designed a strategy to couple biotin with a diethylenetriamine (DETA)-modified PG4.5 dendrimer using click chemistry (Scheme 1). DETA act as a cross-linker to conjugate biotin with the half-generation PAMAM dendrimer (PG4.5) and enable optimum, controlled drug release via proton sponge effect [45,46]. Following the synthesis, the physicochemical and biological properties of gemcitabine loaded PG4.5-DETA-biotin nanoparticles were characterized to examine the therapeutic efficiency of gemcitabine in HeLa cells (Scheme 2).

## 2. Methods 

### 2.1. Chemicals

Biotin, diethylenetriamine (DETA), *N*-hydroxysuccinimide (NHS), dimethyl sulfoxide (DMSO), *N*-ethyl-*N*′-(3-dimethylaminopropyl)carbodiimide (EDC), 4-morpholineethanesulfonic acid (MES), phosphate-buffered saline (PBS), fetal bovine serum (FBS), penicillin/streptomycin, Dulbecco’s modified Eagle’s medium (DMEM), trypsin, trypan blue, and 3-(4,5-dimethylthiazol-2-yl)-2,5-diphenyl tetrazolium bromide (MTT) were purchased from Sigma-Aldrich, Taipei, Taiwan. Aqueous PAMAM generation 4.5 (PG4.5) solution was obtained from Dendritich Inc., Midland, Michigan 48642, USA. Annexin-V Alexa Fluor-488, Annexin V binding buffer, propidium iodide (PI), 4’,6-diamidino-2-phenylindole dihydrochloride (DAPI), and paraformaldehyde were purchased from Thermo-Fisher, Taipei, Taiwan. Cellulose dialysis membrane with a molecular weight cut-off of 6–8 kDa was purchased from CelluSept T1 (Braine-l′ Alleud, Brussels, Belgium). Water was purified using a Milli-Q Plus 185 system to a resistivity higher than 18.2 mΩ/cm. HeLa cells and HaCaT cells were obtained from the Bio Resource Collection and Research Center (Hsinchu, Taiwan).

### 2.2. Instrumentation

Magnetic resonance spectra (NMR) were generated on a Bruker NMR 500 MHz instrument (Bruker Taiwan Co Ltd. Zhubei City, Hsinchu County 302, Taiwan). Fourier transform infrared (FTIR) analysis was performed using a FTIR spectrophotometer (JASCO JASCO International Co., Ltd., Hachioji, Tokyo 192-0046, Japan) in the region of 4000–500 cm^−1^. Raman spectroscopy (CRM 2000; WITec Inc., Ulm, Germany) was used to analyze the Raman shift in the region of 3200 to 500 cm^−1^. A UV–Vis spectrophotometer (Perkin-Elmer Lambda 25 spectrophotometer, Jasco V550 type, Tokyo, Japan) was used to analyze drug-loading and release behavior. Dynamic light scattering (Horiba SZ100 analyzer, Malvern Instruments, Malvern WR14 1XZ, UK) was used to determine the particle size and surface charge of the nanoparticles. An atomic force microscope (NX10, AFM Park System, Suwon, South Korea) was used to analyze structural morphology. An enzyme-linked immunosorbent assay (ELISA) plate reader (Multiskan FC, Thermo Scientific, Winooski, VT, USA) was used to evaluate the cytotoxicity assays. Cell images were captured using a fluorescent microscope (Logos Biosystems, Gyeonggi-do, South Korea). Fluorescence-activated cell sorting (FACS; Becton Dickinson, San Jose, CA 95131, USA) was performed to quantify the cells.

### 2.3. Synthesis of PG4.5-DETA and PG4.5-DETA-biotin

Synthesis of PG4.5-DETA: PG4.5 was modified with DETA followed by biotin using a two-step EDC/NHS reaction (Scheme 1). PG4.5-DETA was synthesized as previously described [25], with slight modifications. Briefly, 30 mg PG4.5 was diluted in 2 mL of 10 mM MES buffer. Five-fold molar excess of EDC (140 mg) and 3-fold molar excess of NHS (50.1 mg) were dissolved in 1 mL of 10 mM MES buffer, added to the PG4.5 solution, and stirred overnight. In another reaction, a 10-fold molar excess of DETA (150 mg) was dispersed in 2 mL of 10 mM MES for 1 h at 0 °C, then activated PG4.5 was added slowly to the DETA solution, adjusted to pH 4.5 and stirred for 2 days. The product was dialyzed against ultrapure water for two days with six exchanges per day, then the purified polymer was lyophilized in a freeze dryer for 2 days.

Synthesis of PG4.5-DETA-biotin: Biotin-conjugated PG4.5-DETA (PG4.5-DETA-biotin) was synthesized by condensing the amine group of PG4.5-DETA and carboxyl group of biotin using the EDC/NHS reaction. Briefly, 24 mg of biotin was dispersed in 10 mL of a dimethyl sulfoxide (DMSO)/water (2:1 v/v) solution. Then, 96 mg of EDC and 58 mg of NHS were added to the biotin solution, stirred vigorously for 3 h and the biotin solution was slowly added into 10 mL of PG4.5-DETA (1 mg/mL) and stirred for 24 h at room temperature. The product was dialyzed against deionized water for 24 h and lyophilized in a freeze dryer for 2 days.

### 2.4. Preparation and Characterization of Gemcitabine-Loaded Nanoparticles

To prepare gemcitabine-loaded nanoparticles, gemcitabine (GEM; 2 mg) was dissolved in DMSO: H_2_O (1:1 v/v) and dispersed at 1 mg/mL in PG4.5-DETA-biotin (10 mL) solution (DMSO: H_2_O, 1:1 v/v) (Scheme 2). The mixture was sonicated for 2 min and stirred for 48 h in the dark at room temperature, and the product was dialyzed against water overnight and lyophilized for 24 h. The amount of gemcitabine loaded into PG4.5-DETA-biotin was quantified by UV–Visible spectroscopy at 268 nm. The drug-loading content (DL %) and encapsulation efficiency (EE %) were calculated against a standard curve using Equations (1) and (2), respectively.
(1)$$\mathrm{DL}\;\left(\%\right)=\frac{\mathrm{Amount}\;\mathrm{of}\;\mathrm{encapsulated}\;\mathrm{gemcitabine}}{\mathrm{Total}\;\mathrm{weight}\;\mathrm{of}\;\mathrm{nanoparticle}}\;\times \;100$$
(2)$$\mathrm{EE}\;\left(\%\right)=\frac{\mathrm{Amount}\;\mathrm{of}\;\mathrm{encapsulated}\;\mathrm{gemcitabine}}{\mathrm{Total}\;\mathrm{gemcitabine}\;\mathrm{added}}\;\times \;100$$

The particle size and surface charge of PG4.5-DETA-biotin/GEM were estimated using dynamic light scattering (DLS). The structural morphology of PG4.5-DETA-biotin/GEM nanoparticles was assessed by AFM.

### 2.5. In Vitro Gemcitabine Release Study

To study the release profile, in vitro release of gemcitabine from biotin-grafted PG4.5-DETA was examined using a dialysis method in PBS at pH 7.4, 6.5, and 5 buffer that mimic the normal physiology, tumor, and endosome microenvironments, respectively. PG4.5-DETA-biotin/GEM solution (1 mg/mL, 1.5 mL) was placed in the dialysis bag (MWCO, 6–8 kDa), immersed in 20 mL of media and shaken at 60 rpm in an incubator at physiological temperature (37 °C). At predetermined time intervals, 3 mL aliquots were drawn and replaced with the same amount of fresh PBS. The cumulative percentage of gemcitabine released was quantified by UV–Visible spectroscopy at 268 nm against a standard curve [47] (Appendix A). 

### 2.6. Cellular Uptake Studies

Cellular uptake of PG4.5-DETA-biotin/GEM nanoparticles was examined using fluorescein isothiocyanate (FITC) as a probe. For qualitative analysis, fluorescence microscopic analysis was used to examine the fluorescence intensity of nanoparticles inside the tumor cells [48]. Briefly, HeLa cells were seeded into confocal dish at 5 × 10^4^ cells density and incubated for 24 h at 37 °C in a 5% CO_2_ environment. Then the media was discarded, the cells were washed, and the cells were incubated in DMEM media (without FBS) containing PG4.5-DETA-biotin/GEM (20 µg/mL of GEM) and excess free biotin for 1 h followed by PG4.5-DETA-biotin/GEM nanoparticles for 2 or 6 h. The cells were washed to remove unabsorbed nanoparticles, incubated with DAPI (300 nM) for 20 min to stain the nuclei and analyzed by fluorescent microscopy.

Flow cytometry was performed to quantify the number of cells that internalized PG4.5-DET-Biotin/GEM nanoparticles. Briefly, HeLa cells were seeded in 6-well plates at 5 × 10^5^ cells/well and incubated for 24 h. The media was replaced with fresh DMEM media (without FBS) containing PG4.5-DETA-biotin/GEM (20 µg/mL of GEM) and excess biotin for 1 h, followed by PG4.5-DETA-biotin/GEM for 6 h, then the cells were washed with PBS, detached with trypsin, centrifuged (5 min, 1000 rpm) and resuspended in 500 µL of PBS. In each independent experiment, a total of 10,000 viable cells were evaluated and FITC fluorescence was analyzed using BD FACSFlow™ software. 

### 2.7. Cytotoxicity and Cell Viability Studies

The biocompatibility of PG4.5-DETA-biotin towards normal and tumor cells and the effects of PG4.5-DETA-biotin/GEM nanoparticles and free gemcitabine on tumor cell proliferation were evaluated using the MTT assay, as previously reported [48]. Briefly, HeLa and HaCaT cells were seeded in 96-well plates at a density of 5 × 10^3^ cells/well in DMEM media and incubated for 24 h at 37 °C. The cells were washed and treated with a range of concentrations of PG4.5-DETA-biotin, PG4.5-DETA-biotin/GEM or free gemcitabine for 24 h. The cells were washed, 100 µL of DMEM media and 5 mg/mL of MTT (20 µL) were added per well, incubated for 4 h at 37 °C, the media was discarded and 100 µL of DMSO was added to each well to dissolve the formazan crystals formed in live cells. Absorbance was measured using a ELISA microplate photometer at 570 nm. Untreated cells were used as a control. Cell viability was calculated as the ratio of the absorbance of treated cells to untreated cells [49].

### 2.8. Apoptosis Assay

Fluorescent activated cell sorting (FACS) assay was performed to analyze apoptotic and necrotic cells using the Annexin V-Alexa Fluor 488 apoptosis kit. Briefly, HeLa cells were seeded in 6-well plates at a density of 1 × 10^6^ cells/well, incubated for 24 h at 37 °C in 5% CO_2_, washed and treated with PBS, free gemcitabine or PG4.5-DETA-biotin/GEM (20 µg/mL gemcitabine) for 24 h. The cells were rinsed with PBS, detached with trypsin, centrifuged (5 min, 1000 rpm), resuspended in 100 µL of Annexin V binding buffer, incubated with Alexa Fluor 488 (5 µL) and PI (1 µL) for 15 min at room temperature according to the manufacturer’s protocol, 400 µL of Annexin V binding buffer was added, and the cells were subjected to FACS to quantify live, apoptotic, and necrotic cells.

### 2.9. Statistical Analysis

Results were expressed as the mean and standard deviation of three independent experiments. Statistical analysis was performed using SPSS package version 22. Differences among groups were analyzed using ANOVA. Statistical significance was considered as *P* < 0.05 (*) and *P* < 0.01 (**). 

## 3. Results and Discussion

### 3.1. Synthesis and Characterization of PG4.5-DETA and PG4.5-DETA-Biotin

The biotin-targeted nanocarrier PG4.5-DETA-biotin was designed to achieve selective delivery of gemcitabine into cancer cells and synthesized via the two-step EDC/NHS coupling reaction shown in Scheme 1. PG4.5-DETA was first modified with DETA, followed by biotin conjugation through amide bond formation. As reported in our previous work, successful conjugation of PG4.5 with DETA was confirmed by ^1^H NMR and FTIR. Similarly, ^1^H NMR was used to confirm successful conjugation of biotin to PG4.5-DETA (Figure 1). New proton peaks appeared at δ 6.4 ppm (a) and 6.3 ppm, (a’), δ 4.2 ppm (b), and 4.3 ppm (b’)**,** and δ 1.2–1.5 ppm (d, e and e’) due to the imidazolidone protons (urea proton), protons adjacent to imidazolidone (methylene proton)**,** and the methylene protons derived from biotin, respectively. The peaks in the range from δ 2.2 to 3.2 ppm (f, h, i, and j) belong to PG4.5-DETA and indicate successful conjugation of PG4.5-DETA to biotin. Integration of the proton peak areas indicated nearly 24% of the primary amine groups were substituted by biotin.

There was no noticeable difference in the band intensities or chemical shifts in the FTIR spectra of PG4.5 and PG4.5-DETA, except for the appearance of a broad peak at 3280 cm^−1^ in PG4.5-DETA due to N-H stretching in the primary amine groups of DETA. The intensity of the bands for the amide-II at 1546 cm^−1^ dramatically increased in PG4.5-DETA-biotin, as illustrated in Figure 2A, probably due to the tetra-imidazole group of biotin. In addition, the slight chemical shift and reduced intensity for the C=O stretching peak confirmed PG4.5-DETA-biotin was successfully synthesized.

The dendrimer modified with DETA and biotin was characterized by Raman spectroscopy (Figure 2B). The appearance of a new Raman band at 1120 cm^−1^ (–CH rock), decreased peak intensity at 885 cm^−1^ (–CH twist), and slight shift of the band around 2850 cm^−1^ (–CH_2_ symmetrical stretch) are most likely due to structural deformation of PG4.5 after conjugation to DETA. In PG4.5-DETA-biotin, the new bands at 820 cm^−1^ (–CN ring bending), 1065 cm^−1^ (S–C stretching), 2849 cm^−1^ and 2889 cm^−1^ (–CH_3_ stretching) belong to biotin, whereas the peaks at 965 cm^−1^, 1300 cm^−1^, 1644 cm^−1^ (amide), and 2933 cm^−1^ (–CH_2_ symmetrical stretching) can be attributed to PG4.5-DETA. The reduction in the Raman shift intensity at 1644 cm^−1^ (primary amine) indicates formation of an amide bond between the primary amines of PG4.5-DETA and the carboxylic group of biotin, further proving successful conjugation of biotin to PG4.5-DETA.

### 3.2. Characterization of Gemcitabine-Loaded Nanoparticles

After biotin was coupled to the DETA-conjugated PG4.5 dendrimer, gemcitabine was encapsulated and the physicochemical characteristics of the drug-loaded nanoparticles were investigated (Scheme 2). Size, surface charge, shape, morphology, and stability are crucial physicochemical features of drug delivery nanocarriers. For instance, nanoparticles between 30 and 200 nm are preferably endocytosed by tumor cells [3]. Production of appropriately sized biotin-coupled PG4.5-DETA dendrimer could enhance the cellular uptake rate via receptor-mediated endocytosis (Scheme 2). As depicted in Table 1, gemcitabine-loaded PG4.5-DETA-biotin formed nanoparticles with a size of 81.6 ± 3.96 nm, slightly smaller than PG4.5-DETA-biotin nanoparticles (82.24 ± 8.49 nm). Hydrogen bonding between gemcitabine and biotin and electrostatic interactions between the drug and unreacted carboxylic groups on the dendrimer may play important roles in the formation of gemcitabine-loaded nanoparticles (Figure 3B). In contrast, PG4.5-DETA-biotin nanoparticles were larger than PG4.5-DETA (49.6 ± 1.95 nm) and pure PG4.5 (3.58 ± 0.49 nm), suggesting that conjugation to biotin increased the diameter of the PG4.5-DETA nanoparticles. The zeta potentials of PG4.5-DETA-biotin and gemcitabine-loaded PG4.5-DETA-biotin nanoparticles were −2.03 ± 0.12 mV and −0.47 ± 1.25 mV, respectively (Table 1); incorporation of gemcitabine into PG4.5-DETA-biotin slightly increased the zeta potential. A nearly neutral surface charge has been shown to promote the formation of stable nanoparticles, associated with non-toxic properties and potentially increases the cellular uptake of nanocarriers into tumor cells [3]. 

The structural morphology of gemcitabine-loaded PG4.5-DETA-biotin nanoparticles was evaluated by AFM and revealed the formation of spherical nanoparticles, as depicted in Figure 3A. Based on these physicochemical findings, biotin-conjugated DETA-modified dendrimer nanoparticles were strongly expected to efficiently deliver drugs into the target tumor cells. 

Drug-loading content and encapsulation efficiency play a fundamental role in drug delivery systems. The loading of gemcitabine into PG4.5-DETA-biotin was evaluated by UV–visible spectroscopy (Figure 4A). The absorption peak for free gemcitabine at 268 nm shifted to 275 nm after encapsulation by PG4.5-DETA-biotin. The drug-loading content and encapsulation efficiency were 10.84 ± 0.16% and 47.01 ± 0.71%, respectively. Previous reports showed that the imidazolidone group of biotin forms hydrogen bonds with the imidazole ring of histidine when adsorbed to proteins during accumulation inside tumor cells [50]. Similarly, hydrogen bonding between the biotin ring and imidazolidone group of gemcitabine (Figure 3B) probably increases the drug-loading capacity [51]. Electrostatic interactions between the drug and the unmodified carboxylate group of PG4.5 may also increase the gemcitabine-loading content of the nanoparticles.

### 3.3. In Vitro Gemcitabine Release Study

Considering the acidic pH of tumor cells and the surrounding tumor microenvironment, pH-dependent gemcitabine release was investigated in vitro at pH 7.4, 6.5, and 5 using a dialysis method at specific time intervals. As illustrated in Figure 4B, a biphasic pattern of gemcitabine release was observed, with rapid release in the first 6 h followed by slow and continuous release over the remaining hours. Approximately 24.5 ± 1.34%, 60.54 ± 1.99%, and 73.96 ± 1.14% of gemcitabine was released after 48 h incubation at pH 7.4, 6.5, and 5, respectively. The lowest percentage release at pH 7.4 indicates the stability of PG4.5-DETA-biotin/GEM nanoparticles under normal physiological conditions. In acidic environments (pH 6.5 and 5) similar to the tumor microenvironment, protonation of secondary amine groups on the dendrimer and imidazoline group of biotin may induce the proton sponge effect and increase the release of gemcitabine [52]. Furthermore, the spontaneous release observed at the initial stage may be due to collapse of the dendrimer surface configuration at acidic pH; collapse of the dendrimer surface configuration has been shown to be responsible for immediate inhibition of tumor cell proliferation [53]. On the other hand, gradual release of gemcitabine due to breakdown of hydrogen bonds may improve the efficacy of gemcitabine.

### 3.4. Cellular Uptake Assay

Effective cellular uptake is an essential prerequisite for a successful drug delivery system. However, the negative charge, small molecular mass, and hydrophilic nature of gemcitabine limit cellular uptake of the drug by cancer cells. Encapsulation of gemcitabine by active tumor-targeted nanoparticles could enhance the cellular internalization and therapeutic effects of this drug, and also reduce the side-effects in normal cells. In this study, gemcitabine was incorporated into a biotin-targeted PAMAM dendrimer to promote uptake by sodium-dependent multivitamin transport (SMVT) receptor-mediated endocytosis [33,42]. Qualitative and quantitative analysis of cellular uptake in HeLa cells were performed using fluorescence microscopy and flow cytometry, respectively. Fluorescently labelled PG4.5-DETA-biotin nanoparticles were prepared by reacting fluorescein isothiocyanate (FITC) dye with the amine groups of DETA-modified PAMAM dendrimer and confirmed by UV–visible spectrophotometry (Appendix A). Free biotin was used to block the biotin receptors to examine the cellular uptake of PG4.5-DETA-biotin/GEM nanoparticles. As illustrated in Figure 5A, fluorescence microscopy revealed strong green fluorescence throughout the cytoplasm of the cells, regardless of the incubation period. Cells incubated with biotin prior to PG4.5-DETA-biotin exhibited lower florescence intensities, indicating blocking the biotin receptors using free biotin reduced cellular uptake of the nanoparticles. On the other hand, cells treated with PG4.5-DETA-biotin exhibited high fluorescence intensities that increased with the incubation period (from 2 to 6 h), demonstrating uptake of the nanoparticles into HeLa cells occurred by vitamin receptor-mediated endocytosis.

The enhanced cellular uptake of PG4.5-DETA-biotin/GEM nanoparticles was also confirmed and quantified by flow cytometry. A higher percentage of cells treated PG4.5-DETA-biotin took up the nanoparticles (81.5%) compared to cells treated with biotin to block the biotin receptors (73.3%) after 6 h incubation (Figure 5B). This result confirmed the fluorescence microscopy analysis, thus we concluded that the biotin-targeted DETA-modified PAMAM dendrimer strongly interacts with the cell membrane of HeLa cells and has an enhanced uptake efficiency through biotin/SMVT receptor-mediated endocytosis. 

### 3.5. Cytotoxicity and Cell Viability Studies

The cytotoxicity of PG4.5-DETA-biotin towards normal HaCaT cells and HeLa tumor cells was evaluated using the MTT assay. As illustrated in Figure 6A, PG4.5-DETA-biotin did not exert significant toxicity at high concentrations (500 µg/mL), indicating the biotin-targeted dendrimer was biocompatible and did not affect the viability of normal or tumor cells. Next, we examined the effect of free GEM and PG4.5-DETA-biotin/GEM on the viability of HeLa cells and observed significant reductions in cell viability (*P* < 0.05 at 20 µg/mL GEM or higher concentrations) compared to control cells (Figure 6B). The cytotoxicity of PG4.5-DETA-biotin/GEM increased with the drug concentration (1.35, 2.7, 5.4, 10.8, 21.6, 32.4, and 43.2 μg/mL gemcitabine). The anti-proliferative activity of free GEM was slightly higher than that of PG4.5-DETA-biotin/GEM, possibly to due to the smaller molecular size of free GEM enabling more rapid cellular uptake and transport into the nucleus to inhibit proliferation. Therefore, incomplete release of encapsulated gemcitabine from the nanoparticles, as observed in the previous release experiment could reduce the anti-proliferative effects of the drug.

### 3.6. Apoptosis Assays

Induction of apoptosis and necrosis in cancer cells are crucial, direct methods of destroying tumors [54]. In the present study, the numbers of necrotic and apoptotic HeLa cells were assessed by Alexa^®^ Fluor 488 Annexin-V and PI double staining. As illustrated in Figure 7, the FACS analysis revealed 97.1% of control HeLa cells were viable. After treatment with free gemcitabine, biotin + PG4.5-DETA-biotin/GEM, and PG4.5-DETA-biotin/GEM, 93.9%, 53.2%, and 78.9% of the cells were apoptotic, respectively, indicating PG4.5-DETA-biotin/GEM nanoparticles could be used to effectively eliminate viable tumor cells. Moreover, 4.5%, 1.3%, and 18.4% of the cells treated with free GEM, biotin + PG4.5-DETA-biotin/GEM, and PG4.5-DETA-biotin/GEM were necrotic, respectively. Thus, the apoptosis assays confirmed the anti-proliferative activity of PG4.5-DETA-biotin/GEM nanoparticles observed in the MTT assay and demonstrated that PG4.5-DETA-biotin/GEM nanoparticles significantly induced apoptosis and necrosis compared to biotin receptor-blocked cells, most likely due to the enhanced cellular uptake efficiency of PG4.5-DETA-biotin/GEM via SMVT receptor-mediated endocytosis. Therefore, this biotin-targeted drug delivery system can effectively induce apoptosis in tumor cells in vitro.

## 4. Conclusions 

In the present study, the PAMAM dendrimer was modified with DETA followed by conjugation to biotin through the amide bond via EDC/NHS chemistry to develop an active biotin-targeted drug nanocarrier. Our ^1^H NMR, FTIR, and Raman spectroscopy analysis confirmed the coupling of DETA and biotin to the dendrimer. Gemcitabine was loaded to form PG4.5-DETA-biotin/GEM nanoparticles, which exhibited a spherical shape with a diameter of 81.60 ± 6.08 nm and 0.47 ± 1.25 mV surface charge. The drug-loading content and encapsulation efficiency were 10.84 ± 1.06% and 47.01 ± 0.71%, respectively. Flow cytometry and fluorescent microscopy demonstrated the PG4.5-DETA-biotin/GEM nanoparticles exhibited high cellular uptake efficiency, indicating the nanoparticles precisely targeted the cancer cells and were taken up via SMVT mediated endocytosis. The in vitro cytotoxicity assays revealed that PG4.5-DETA-biotin was non-toxic whereas PG4.5-DETA-biotin/GEM nanoparticles exerted high cytotoxicity and anti-proliferative activity in HeLa cells. Furthermore, double staining and FACS analysis proved PG4.5-DETA-biotin/GEM nanoparticles induced significant apoptosis. Based on these results, we conclude that PG4.5-DETA-biotin nanoparticles could represent a potential vitamin-targeted nanocarrier system for tumor-specific biotin-targeted delivery of anticancer drugs. Further studies are in progress to develop these nanoparticles as an in vivo delivery system.

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
