# Peer review of "Biotin-Decorated PAMAM G4.5 Dendrimer Nanoparticles to Enhance the Delivery, Anti-Proliferative, and Apoptotic Effects of Chemotherapeutic Drug in Cancer Cells"

_pharmaceutics, 2020, doi:10.3390/pharmaceutics12050443_

Round 1

Reviewer 1 Report

This work experimentally shows that the dendrimer nanoparticles chemically modified with biotin molecules could be considered as a potential anticancer drug delivery system which aims at decreasing the acute side effects of chemotherapy by means of targeting the biotin receptors overexpressed on tumor cells. The paper is well written and I recommend it for publication after minor revisions:

  1. In order to estimate the potential of the proposed approach, it would be beneficial for the readers of Pharmaceutics, if the authors discuss how much quantitatively the functional and molecular expression of sodium dependent multivitamin transporter (SMVT) is lower on normal cells compared to cancer cells.
  2. Is there any reasoning on the author’s choice of gemcitabine as a cargo for delivery in comparison with other existing chemotherapeutic agents? It would be nice to discuss this issue in Introduction.
  3. It is strange that cytotoxicity of unloaded nanoparticles was checked both on cancer and normal cells (Fig. 6A), but only on cancer cells for GEM-loaded particles (Fig. 6B). Comparison with the normal cell viability would allow one to figure out how much the side effects could be diminished as a result of using the biotin receptor-targeted delivery.
  4. In Fig. 6A, some cell viabilities are greater than 100%. Also, there is a noticeable effect of the biotin-modified nanoparticles on the viability of the tumor cells. It looks like biotin conjugated molecules are showing higher cytotoxicity in biotin positive cancer cells than in the normal cell. Those observations must be properly addressed.
  5. Actually, AFM picture (Fig. 3A) is too blurry to be informative. I think no one could see “smooth-surfaced, uniformly distributed spherical particles” and definitely, this figure does not give a hint that these particles “were strongly expected to efficiently deliver drugs into the target tumor cells.”
  6. Since the uptake of nanocarriers into the cell as well as the release from them are both pH-dependent, the pH must be controlled or at least specified for Cellular Uptake Assay.
  7. In Figure 5, it would be reasonable to compare the affinities of biotin-conjugated particles uptake in cancer and normal cells.
  8. The paper needs some editing in terms of “typos” (for example, there is no Fig. 8, see p.21) and formatting.

Author Response

Authors Response for Reviewer – 1:

Point 1: In order to estimate the potential of the proposed approach, it would be beneficial for the readers of Pharmaceutics, if the authors discuss how much quantitatively the functional and molecular expression of sodium dependent multivitamin transporter (SMVT) is lower on normal cells compared to cancer cells.

Authors Response: Sodium dependent multivitamin transporter (SMVT) mediate the internalization of biotin into cells. SMVT consists of 635 amino acids with 12 transmembrane domains located on chromosome 2p23[1]. The regulation of histone biotinylation were used to identify the molecular expression of SMVT gene in human Jurkat lymphoblastoma cells[2]. SMVT has low affinity and concentration dependent activity. It appears to be coupled to electrochemical gradient of sodium but not to proton gradient. The transporter became saturated at 10 µM biotin concentration in normal cells. But, the affinity in the cells and tissues depend on their origin, nature of the cell line and disease condition. Several tissues such as intestine, placenta, liver, cornea and kidney express SMVT[3]. The overexpression of SMVT is much higher in several lung, renal, colon, intestinal, breast and ovarian cancer cells[4,5].

Point 2: Is there any reasoning on the author’s choice of gemcitabine as a cargo for delivery in comparison with other existing chemotherapeutic agents? It would be nice to discuss this issue in Introduction.

Authors Response: In our design, gemcitabine was exceptionally selected among the other chemotherapeutic drugs and loaded into our nanocarrier to improve its efficiency. In spite of their common characteristics among chemotherapeutic drugs, gemcitabine has unique pharmacological properties and advantage such as its potent cytostatic and cytotoxic activity along with manageable toxicities [6]. It has also a unique and multiple intracellular targets for drug–molecular interactions effect in cancer tissue. Unlike other chemotherapeutic drugs to inhibit polymerase enzymes, gemcitabine can incorporate itself into DNA by replacing cytidine to trigger anti-proliferative activity [7]. Furthermore, it is a gold-standard chemotherapeutic drug for unfit and elderly patient of advanced cancer[8]. Unfortunately, it is vulnerable for enzymatic degradation and rapidly deaminated due to its lower molecular weight and hydrophilic properties. Hence, we have chosen gemcitabine to deliver by smart nanocarrier that could mitigate the main challenges and improve the efficiency.

And, this comment is now incorporated in the introduction part manuscript. Page -3, line 4-9.

Point 3: It is strange that cytotoxicity of unloaded nanoparticles was checked both on cancer and normal cells (Fig. 6A), but only on cancer cells for GEM-loaded particles (Fig. 6B). Comparison with the normal cell viability would allow one to figure out how much the side effects could be diminished as a result of using the biotin receptor-targeted delivery.

Authors Response: Authors appreciate the comments on cytotoxicity of gemcitabine loaded nanocarrier in normal cells. The cytotoxicity of drug loaded nanocarrier is basically related with the releasing profile of drug in the cell[9]. In our study, we designed a pH dependent drug releasing mechanism. Normal and cancer cell have pH 7.4 and 6.5-6.8, respectively[10]. By considering these pH values, the releasing profile of gemcitabine at pH 6.5 (to mimic cancer cell) was meaningfully higher than pH 7.4 (mimic normal cell) (Fig.4B). The releasing profile was not good enough at normal pH rather it showed the stability of drug loaded nanoparticle at normal pH. The weak drug releasing profile at normal cell (pH~7.4) could not lead to a significant cytotoxicity. Thus, we decided to escape the cytotoxic effect of gemcitabine loaded nanocarrier in normal cells. In addition, Fig.6A (page-23) represent the biocompatibility result of PG4.5-DETA-Biotin nanoparticles in both normal and cancer cells. It used to confirm whether PG4.5-DETA-Biotin nanoparticle is toxic or not in both normal and cancer cells. Thus, Fig. 6.A can serve as backbone to explain and confirm that the cytotoxic properties of gemcitabine loaded nanocarrier (Fig.6B) were due to the effect of drug only. That is why in Fig. 6B (page-23), we focused on cancer cells to enhance the antiproliferative activity and trigger apoptosis induction of gemcitabine by loading into PG4.5-DETA-Biotin. Furthermore, biotin here in this study was used only for targeting purpose to enhance the cellular internalization. We strongly recognize the comment to investigate the role of biotin receptor on cytotoxicity and considered it as input for the future work. 

Point 4: In Fig. 6A, some cell viabilities are greater than 100%. Also, there is a noticeable effect of the biotin-modified nanoparticles on the viability of the tumor cells. It looks like biotin conjugated molecules are showing higher cytotoxicity in biotin positive cancer cells than in the normal cell. Those observations must be properly addressed.

Author Response: After PG4.5-DETA-Biotin treatment, the cells have access for micronutrient biotin that promote cell division and growth in both normal and cancer cells. Hence, the cell viability of treated cells could be higher than untreated cells (control group) especially at lower concentration. These couldn’t be the same at higher concentration. There was slight change in cytotoxicity in normal and cancer cell, the difference between them weren’t significant. The slight increment of cytotoxicity in cancer cells as compared to normal cell could be the acidic properties of the cells. The extracellular compartment of cancer is acidic (pH~6.5 to 6.8)[10]. In acidic environment, the imidazole group of biotin and amine groups of modified dendrimer can undergo protonation and become positive surface charged nanocarrier [11,12]. These positive surface charged nanocarrier then easily disrupt the membrane that might cause cytotoxicity [13]. Having this principle, our nanocarrier might cause slight difference in cytotoxicity between cancer and normal cells. 

Point 5: Actually, AFM picture (Fig. 3A) is too blurry to be informative. I think no one could see “smooth-surfaced, uniformly distributed spherical particles” and definitely, this figure does not give a hint that these particles “were strongly expected to efficiently deliver drugs into the target tumor cells”

Author Response: We appreciate the reviews comment on the morphology of nanoparticles. We modified the explanations to show only formation of spherical nanoparticles. But, the concept and statement “were strongly expected to efficiently deliver drugs into the target tumor cells” is misinterpreted and the full statement is “Based on these physicochemical findings, biotin-conjugated DETA-modified dendrimer nanoparticles were strongly expected to efficiently deliver drugs into the target tumor cells” Page -17, line -3. It is written to express not only AFM result but also other physicochemical analysis including hydrodynamic diameter, surface charge and loading and encapsulation efficiency. For instance, the DLS result revealed the appropriate particle size (81.60 ± 6.08 nm) and surface charge (0.47 ± 1.25) for effective delivery of drugs into the target tumor cells[6,14].

Point 6: Since the uptake of nanocarriers into the cell as well as the release from them are both pH-dependent, the pH must be controlled or at least specified for Cellular Uptake Assay.

Authors Response: We designed sodium dependent multivitamin transporter (SMVT) mediated endocytosis for the internalization of biotin decorated PAMAM dendrimer. The transporter are located at the surface of cancer cells to recognize the nanocarrier. Successful cellular uptake requires a stable nanocarrier. We tried our best to prepare a stable nanocarrier at different pH value in PBS and finally achieved at pH 7.4. And, it was difficult to obtain a stable nanocarrier at lower pH like pH ~6.5-6.8 to mimic cancer cell and pH~5 to mimic endosome compartment due to repulsion of protonated groups and aggregation of nanocarrier. Thus, we maintain pH to be 7.4 during the preparation of nanocarrier for cellular uptake application. In addition, it couldn’t be easy to control pH inside the cells and correlate the cellular uptake with releasing behaviors. Cellular uptake is carried out on surface of the cell but releasing is managed in the intracellular compartment with different pH range. The in vitro experiment showed a distinct releasing profiles at pH 7.4, 6.4 and 5 that used to predict pH regulation can achieve in vitro. However, there is no a scientific approach to control pH for cellular uptake in the live cell rather we adjusted pH7.4 during nanocarrier preparation. 

Point 7: In Figure 5, it would be reasonable to compare the affinities of biotin-conjugated particles uptake in cancer and normal cells.

Authors Response: Authors appreciate the comment to compare cellular internalization of biotin conjugated nanoparticles in both normal and cancer cells. Cells express biotin receptor to fulfill the biotin demand for biosynthesis of carbohydrates, short chain amino acids and fatty acids and promotion of cell growth. Biotin demand in cancer cell is significantly higher than normal cells. Hence, cancer cells have higher affinity for biotin and biotin-conjugated nanocarrier in order to meet biotin demand for their extensive metabolic activity [15]. Like free biotin, biotin-conjugated nanoparticles can be internalized but the efficiency, quantity and affinity might be depend on the number of expressed receptors and incubation period. More importantly, the pKa of biotin is nearly 4–5, a higher biotin uptake is observed in lower pH microenvironment of cancer cells[3]. Previously, several papers have been reported on the affinity of biotin-conjugated nanoparticles in normal cells[16,17]. Hence, there is no scientific ground to do and approve it again.

Point 8: The paper needs some editing in terms of “typos” (for example, there is no Fig. 8, see p.21) and formatting.

Authors Response: The font side and other editing were made including Fig. 8 which was Fig.7. Page-24 and line-6.

Reference

  1. Yellepeddi, V.K.; Kumar, A.; Maher, D.M.; Chauhan, S.C.; Vangara, K.K.; Palakurthi, S. Biotinylated PAMAM Dendrimers for Intracellular Delivery of Cisplatin to Ovarian Cancer: Role of SMVT. 2011, 31, 897-906.
  2. Vadlapudi, A.D.; Vadlapatla, R.K.; Pal, D.; Mitra, A.K. Biotin uptake by T47D breast cancer cells: Functional and molecular evidence of sodium-dependent multivitamin transporter (SMVT). International Journal of Pharmaceutics 2013, 441, 535-543, doi:https://doi.org/10.1016/j.ijpharm.2012.10.047.
  3. Vadlapudi, A.D.; Vadlapatla, R.K.; Mitra, A.K. Sodium dependent multivitamin transporter (SMVT): a potential target for drug delivery. Curr Drug Targets 2012, 13, 994-1003, doi:10.2174/138945012800675650.
  4. Kou, L.; Bhutia, Y.D.; Yao, Q.; He, Z.; Sun, J.; Ganapathy, V. Transporter-Guided Delivery of Nanoparticles to Improve Drug Permeation across Cellular Barriers and Drug Exposure to Selective Cell Types. 2018, 9, doi:10.3389/fphar.2018.00027.
  5. Kou, L.; Bhutia, Y.D.; Yao, Q.; He, Z.; Sun, J.; Ganapathy, V. Transporter-Guided Delivery of Nanoparticles to Improve Drug Permeation across Cellular Barriers and Drug Exposure to Selective Cell Types. Front Pharmacol 2018, 9, 27, doi:10.3389/fphar.2018.00027.
  6. Arya, G.; Vandana, M.; Acharya, S.; Sahoo, S.K. Enhanced antiproliferative activity of Herceptin (HER2)-conjugated gemcitabine-loaded chitosan nanoparticle in pancreatic cancer therapy. Nanomedicine 2011, 7, 859-870, doi:10.1016/j.nano.2011.03.009.
  7. Derakhshandeh, K.; Fathi, S. Role of chitosan nanoparticles in the oral absorption of Gemcitabine. Int J Pharm 2012, 437, 172-177, doi:10.1016/j.ijpharm.2012.08.008.
  8. Toschi, L.; Finocchiaro, G.; Bartolini, S.; Gioia, V.; Cappuzzo, F. Role of gemcitabine in cancer therapy. Future Oncology 2005, 1, 7-17, doi:10.1517/14796694.1.1.7.
  9. Hossen, S.; Hossain, M.K.; Basher, M.K.; Mia, M.N.H.; Rahman, M.T.; Uddin, M.J. Smart nanocarrier-based drug delivery systems for cancer therapy and toxicity studies: A review. Journal of Advanced Research 2019, 15, 1-18, doi:https://doi.org/10.1016/j.jare.2018.06.005.
  10. Hao, G.; Xu, Z.P.; Li, L. Manipulating extracellular tumour pH: an effective target for cancer therapy. RSC Advances 2018, 8, 22182-22192, doi:10.1039/C8RA02095G.
  11. Maiti, S.; Paira, P. Biotin conjugated organic molecules and proteins for cancer therapy: A review. European Journal of Medicinal Chemistry 2018, 145, 206-223, doi:https://doi.org/10.1016/j.ejmech.2018.01.001.
  12. Kanayama, N.; Fukushima, S.; Nishiyama, N.; Itaka, K.; Jang, W.D.; Miyata, K.; Yamasaki, Y.; Chung, U.I.; Kataoka, K. A PEG-based biocompatible block catiomer with high buffering capacity for the construction of polyplex micelles showing efficient gene transfer toward primary cells. ChemMedChem 2006, 1, 439-444, doi:10.1002/cmdc.200600008.
  13. Zhang, C.; An, T.; Wang, D.; Wan, G.; Zhang, M.; Wang, H.; Zhang, S.; Li, R.; Yang, X.; Wang, Y. Stepwise pH-responsive nanoparticles containing charge-reversible pullulan-based shells and poly(beta-amino ester)/poly(lactic-co-glycolic acid) cores as carriers of anticancer drugs for combination therapy on hepatocellular carcinoma. J Control Release 2016, 226, 193-204, doi:10.1016/j.jconrel.2016.02.030.
  14. Wang, D.; Liang, N.; Kawashima, Y.; Cui, F.; Yan, P.; Sun, S. Biotin-modified bovine serum albumin nanoparticles as a potential drug delivery system for paclitaxel. Journal of Materials Science 2019, 54, 8613-8626, doi:10.1007/s10853-019-03486-9.
  15. Rompicharla, S.V.K.; Kumari, P.; Bhatt, H.; Ghosh, B.; Biswas, S. Biotin functionalized PEGylated poly(amidoamine) dendrimer conjugate for active targeting of paclitaxel in cancer. Int J Pharm 2019, 557, 329-341, doi:10.1016/j.ijpharm.2018.12.069.
  16. Uram, L.; Filipowicz, A.; Misiorek, M.; Pienkowska, N.; Markowicz, J.; Walajtys-Rode, E.; Wolowiec, S. Biotinylated PAMAM G3 dendrimer conjugated with celecoxib and/or Fmoc-l-Leucine and its cytotoxicity for normal and cancer human cell lines. Eur J Pharm Sci 2018, 124, 1-9, doi:10.1016/j.ejps.2018.08.019.
  17. Cheng, M.; Ma, D.; Zhi, K.; Liu, B.; Zhu, W. Synthesis of Biotin-Modified Galactosylated Chitosan Nanoparticles and Their Characteristics. Cellular Physiology and Biochemistry 2018, 50, 569-584, doi:10.1159/000494169.

Reviewer 2 Report

The subject matter of submitted manuscript is relative to the subjects of Pharmaceutics. The title does adequately describe the contents of the paper and the abstract is informative enough. In this study, authors dealt with preparation, characterization and evaluation of PAMAM G4.5 Dendrimer nanoparticles for delivery of chemotherapeutic drug in cancer cells. Analysis and discussion of experimental data are well documented.  

The submitted manuscript in the present form is suitable for publication in the Pharmaceutics and it will be further improved after some additions/corrections.

  • What is the number of repetitions in the experiments for the determination of drug-loading content (DL%) and encapsulation efficiency (EE%)?
  • What is the mechanism of gemcitabine release from PG4.4-DETA-Biotin/GEM nanoparticles? The kinetic analysis of drug release during release experiments would provide significant information relative to the mechanism of drug release from tested nanoparticles. There is no reference to the submitted manuscript for kinetic analysis of experimental data.

Author Response

Authors Response for Reviewer-2:

Point 1: What is the number of repetitions in the experiments for the determination of drug-loading content (DL %) and encapsulation efficiency (EE %)?

Authors Response: Drug loading content (DL %) and encapsulation efficiency (EE %) were analyzed in triplicate experiments.

Point 2: What is the mechanism of gemcitabine release from PG4.5-DETA-Biotin/GEM nanoparticles? The kinetic analysis of drug release during release experiments would provide significant information relative to the mechanism of drug release from tested nanoparticles. There is no reference to the submitted manuscript for kinetic analysis of experimental data.

Authors Response: The authors extremely appreciate this important questions on the gemcitabine releasing profile. The releasing experiments were performed by dialysis method in PBS medium. Having the calibration curve, the cumulative releasing kinetics were estimated by the following mathematical formulation[1]:

Cumulative percentage Release (%) = (3 ml/20 ml) x P (t-1) + Pt, where 3 ml represents the volume of sample withdrawn and 20 ml represents volume of dialysate.

The releasing profile revealed biphasic pattern of release gemcitabine i.e. burst release for the first few hours at the surface of nanocarrier and then sustain release for the remaining hours. Thus, the kinetic analysis was more fitted with the Fick’s second law, Korsmeyer– Peppas and Higuchi model indicating diffusion- controlled mechanism of gemcitabine release[2].

We also cited a reference for releasing experiment in manuscript. Page - 10 , line -  18. 

For further information on calculation, Please see the attachment!!!

Reference

  1. Chandrasekaran, A.R.; Jia, C.Y.; Theng, C.S.; Muniandy, T.; Muralidharan, S.; So, D. Invitro studies and evaluation of metformin marketed tablets-Malaysia. Journal of Applied Pharmaceutical Science 2011, 1, 214-217.
  2. Wojcik-Pastuszka, D.; Krzak, J.; Macikowski, B.; Berkowski, R.; Osiński, B.; Musiał, W. Evaluation of the Release Kinetics of a Pharmacologically Active Substance from Model Intra-Articular Implants Replacing the Cruciate Ligaments of the Knee. Materials (Basel) 2019, 12, 1202, doi:10.3390/ma12081202.

Reviewer 3 Report

The manuscript entitled: “Biotin-decorated PAMAM G4.5 Dendrimer Nanoparticles to Enhance the Delivery, Anti-Proliferative and Apoptotic Effects of Chemotherapeutic Drug in Cancer Cells” is a very interesting and complete study about the preparation, physicochemical characterization and biodistribution of gemcitabine-loaded Biotin-DETA-PG4.5. I recommend the publication of the manuscript after elucidate some details:

- I recommend the authors to add the key-words: “biotin” and “gemcitabine”;

- I recommend the authors to determine the polydispersity índex of the nanosystems in the absence and in the presence of the drug by Dynamic Light Scattering (DLS), in order to evaluate the stability and dispersity of the prepared nanosystems;

- I recommend the authors to standardize the font size throughout the article and improve the english of the manuscript.

Author Response

Point 1: - I recommend the authors to add the key-words: “biotin” and “gemcitabine”;

Authors Response: The author appreciate the comments to add “biotin” and “gemcitabine”. These words are now added by deleting the phrase “first phase metabolism” from key-words list. Page -2, Line-21

Point 2: - I recommend the authors to determine the polydispersity índex of the nanosystems in the absence and in the presence of the drug by Dynamic Light Scattering (DLS), in order to evaluate the stability and dispersity of the prepared nanosystems;

Authors Response: PDI values were already documented in the manuscript on table-1, page- 16, column -3. The PDI values were less than 0.5 that indicate the stability and monodispersity of nanoparticles in the presence and absence gemcitabine. 

Point 3: - I recommend the authors to standardize the font size throughout the article and improve the english of the manuscript.

Authors Response: We recognize the reviewers comment to have a uniform and standardize font size throughout the manuscript. Now, we checked, made it uniform and standardized and revised language for whole manuscript.